# Ionic Liquid-Based Polyoxometalate Incorporated at ZIF-8: A Sustainable Catalyst to Combine Desulfurization and Denitrogenation Processes

**DOI:** 10.3390/molecules27051711

**Published:** 2022-03-05

**Authors:** Dinis F. Silva, Alexandre M. Viana, Isabel Santos-Vieira, Salete S. Balula, Luís Cunha-Silva

**Affiliations:** 1LAQV/REQUIMTE, Department of Chemistry and Biochemistry, Faculty of Sciences, University of Porto, 4169-007 Porto, Portugal; josedinis.silva@fc.up.pt (D.F.S.); up201405091@fc.up.pt (A.M.V.); 2CICECO—Aveiro Institute of Materials, Department of Chemistry, University of Aveiro, Campus Universitário de Santiago, 3810-193 Aveiro, Portugal; ivieira@ua.pt

**Keywords:** zeolitic imidazolate frameworks, polyoxomolybdate, desulfurization, oxidative catalysis, ionic liquid, hydrogen peroxide, 1-Butyl-3-methylimidazolium cation

## Abstract

An effective and sustainable process capable of simultaneously execute desulfurization and denitrogenation of fuels is in fact an actual necessity in the refinery industry. The key to achieve this goal is the parallel oxidation of sulfur and nitrogen compounds present in fuels, which is only achieved by an active and recovered catalyst. A novel heterogeneous catalyst was successfully prepared by the encapsulation of an imidazolium-based polyoxometalate (POM) into a ZIF-8 framework ([BMIM]PMo_12_@ZIF-8). This composite material revealed exceptional catalytic efficiency to concurrently proceed with the oxidative desulfurization and denitrogenation of a multicomponent model fuel containing various sulfur and nitrogen compounds. A complete removal of all these compounds was achieved after only one hour and the catalyst system was able to be reused for ten consecutive cycles without loss of efficiency. In fact, an ionic liquid POM was incorporated in the ZIF-8 for the first time, and this composite compound was originally applied as a catalyst for simultaneous oxidative desulfurization and denitrogenation processes.

## 1. Introduction

Fuel consumption and combustion is known to cause emissions of SO_x_ and NO_x_ which are harmful to the environment [1,2]. Consequently, regulations have been put in place to limit the amount of sulfur and nitrogen present in commercial fuels. To meet these parameters, the fuel industry resorts to hydrotreating processes such as hydrodesulfurization (HDS) and hydrodenitrogenation (HDN) [3]. However, these techniques are considerably unsustainable since they demand high economic and energetic costs, and they require high temperature (>350 °C) and pressure (up to 6MPa) conditions [3,4]. While N-based compounds are the most difficult to extract by HDN, the removal by HDS of some S-based compounds, such as benzothiophene derivatives, has also proven to be a difficult task [3,4,5].

Oxidative desulfurization (ODS) and oxidative denitrogenation (ODN) have been presented as low-cost, eco-friendly and efficient alternatives to HDS and HDN, respectively [6,7,8,9]. These processes occur under lower temperatures, at atmospheric pressure and with the aid of a green oxidant and a suitable catalyst [10,11,12]. Currently, one of the challenges associated with the scale-up of ODS/ODN processes relates to catalysts durability and reutilization [13,14].

Polyoxometalates (POMs) are molecular clusters of transition metal oxide anions, with their negative charge being balanced by counter-cations. Keggin-type heteropolyoxometalates [X^n+^M_12_O_40_]^(8 − n)−^ have a central heteroatom (X e.g., B, Si, P) that provides a higher structural stabilization, thermal stability and reduction-oxidation properties [15,16].

Keggin POMs are known for their potential of application in homogeneous oxidative catalytic systems, such as ODS and ODN [15,17,18,19]. One of the approaches employed to overcome the usual drawbacks associated with homogeneous catalysis, like poor recyclability potential, is their encapsulation or incorporation in a suitable porous material.

Zeolitic imidazolate frameworks (ZIFs) are a subclass of Metal-Organic Frameworks (MOFs) that have been studied and described as highly porous and crystalline materials with a wide range of potential application [20,21]. ZIFs have a 3D structure with imidazolate ligands linked with tetrahedral transition metal cations, such as Zn^2+^ and Co^2+^, to give diverse frameworks with interesting mechanical, chemical and thermal stability, good biocompatibility and tuneable pore environments [22,23]. ZIF-8 is a well-known robust structure characterized by its high surface area (S around 1410 m^2^ g^−1^) and pore volume (V around 0.546 cm^3^ g^−1^) [24,25,26]. These properties allow ZIF-8 to be a promising host structure for the heterogenization of homogeneous catalysts like Keggin POMs.

In previous works from our research group, several ionic liquid phosphomolybdates revealed to be efficient homogeneous catalysts for ODS/ODN processes [17,27,28]. Thus, following our scientific and technological interest in the development of POM@MOF-based catalysts, [29,30,31,32,33,34,35,36,37,38,39] the 1-Butyl-3-methylimidazolium phosphomolybdate, [BMIM]_3_PMo_12_O_4_, was incorporated in a porous ZIF-8 support to prepare a robust heterogeneous catalyst suitable for simultaneous ODS/ODN catalytic systems, [BMIM]PMo_12_@ZIF-8 (Figure 1).

## 2. Results and Discussion

### 2.1. Materials Characterization

ZIF-8 and derived [BMIM]_3_PMo_12_@ZIF-8 composite material were prepared through a straightforward room temperature procedure adapted from a synthesis method described elsewhere [24]. The obtained solid compounds were isolated, dried, and characterized by powder X-ray diffraction (PXRD), Fourier-transform infrared (FT-IR) spectroscopy, as well as scanning electron microscopy with electron dispersive X-ray spectroscopy (SEM/EDS). The PXRD patterns obtained for the prepared pristine MOF sample ZIF-8 (Figure 1a) agree with previously reported diffractograms for this crystalline structure, showing all characteristic diffraction peaks with expected relative intensities [40]. [BMIM]_3_PMo_12_@ZIF-8 revealed an equivalent single-phase diffraction pattern which points to the successful formation of the MOF as a host framework. Infrared spectra of both samples show every absorption band expected from ZIF-8 (Figure 1b): weak and strong (C‒N) ring bond vibration bands are observable from 600 up to 1500 cm^−1^, weak (C=N) bond vibration bands can be seen around 1580 cm^−1^, and very strong (Zn‒N) coordination bond vibration bands are registered around 420 cm^−1^ [41]. In the acquired spectra for [BMIM]_3_PMo_12_@ZIF-8, a group of additional medium intensity absorption bands are clearly observable in the interval from 775 to 975 cm^−1^. These can be attributed to different (Mo=O) and (Mo‒O) bond vibrations associated with the encapsulation of the POM in the MOF vacancies [28]. Also, a weak absorption band can be seen around 1080 cm^−1^ which can be related to a (P‒O) bond vibration. 

SEM micrographs obtained for pristine ZIF-8 (Figure 2a) show aggregated particles with regular rhombic dodecahedron morphology and uniform size distribution around 100 nm. For [BMIM]_3_PMo_12_@ZIF-8, similar sized particles with decreased degree of morphological regularity and apparent increased degree of aggregation are observable (Figure 2b). In these, the uniform distribution of Mo is revealed by EDS analysis, suggesting good dispersion of [BMIM]_3_PMo_12_ on the ZIF-8 support. 

### 2.2. Catalytic Desulfurization/Denitrogenation Studies

Combined ODS/ODN catalytic experiments were performed under in a biphasic system using 1-butyl-3-methylimidazole hexafluorophosphate ([BMIM]PF_6_) and multicomponent S and N model fuel. The efficiency of desulfurization and denitrogenation was determined by the periodical analysis of a multicomponent model fuel phase. Pristine ZIF-8 showed no intrinsic catalytic activity towards either the ODS or ODN processes.

[BMIM]_3_PMo_12_@ZIF-8 was applied as a heterogeneous catalyst in combined ODS/ODN processes, and its catalytic behaviour was compared to the homogeneous active center [BMIM]_3_PMo_12_ (Figure 3). It was observed for both catalytic systems, besides an initial propensity towards ODN aided by a superior extraction step and a higher catalytic efficiency towards ODS. After the first hour of the combined processes, near complete sulfur and nitrogen removal were achieved (Figure 3a). Homogeneous and heterogeneous catalysts used to treat the same model fuel under the same experimental conditions (Figure 3b) presented similar catalytic efficiency, since similar profiles for sulfur and nitrogen compounds were obtained. In both homogeneous and heterogeneous catalytic systems, the removal rates of each compound present in the multicomponent model fuel follow the order: IND > QUI > DBT > MDBT > DMDBT ~ BT. Identical results were obtained and previously discussed by our research group using homogeneous phosphomolybate catalysts incorporating different organic cations [27].

The heterogeneous [BMIM]_3_PMo_12_@ZIF-8 catalyst and the ionic liquid extraction phase ([BMIM]PF_6_) were reused for ten consecutive combined ODS/ODN cycles (Figure 4). This procedure allowed the sustainability and increased the cost-efficiency of the fuel treatment process. It is possible to observe that the combined ODS/ODN efficiency is maintained or even slightly increased after the fifth cycle. This increases after the fifth cycle is promoted by the increases of ODS. In fact, a near total desulfurization of the model fuel was observed from the fifth to the tenth cycle (always higher than 98.5%), in detriment of denitrogenation (which stabilizes around 82%). Therefore, these results confirm the potential of the reutilization of the [BMIM]_3_PMo_12_@ZIF-8/[BMIM]PF_6_ system combining the ODS and the ODN processes. Recently, our research group used a heterogeneous catalyst having the phosphomolybdic acid (H_3_PMo_12_) as an active center immobilized into porous MOF-808 support (H_3_PMo_12_@MOF-808) [30]. Similar results were obtained for simultaneous ODS and ODN processes (near complete desulfurization and denitrogenation achieved after 1 h); however, using the H_3_PMo_12_@MOF-808 catalyst, this presented a loss of activity for the ODS process after the fourth cycle. The superior maintenance of activity in reusing ODS/ODN processes with [BMIM]_3_PMo_12_@ZIF-8 catalyst can be related to the structural properties of MOF support (pore and windows size and/or metal center nature). Other studies in the literature using PMo_12_@MOFs catalysts only performed a single ODS process, needing higher quantities of catalysts or using other less sustainable oxidants [13,42,43].

The stability of the catalyst after the first and tenth cycles was investigated by analysing the recovered solid by PXRD and SEM/EDS (after being washed and dried). Furthermore, the resultant model fuel and the extraction phases were analysed by ^31^P NMR to verify the occurrence of possible PMo_12_ leaching. The post-catalytic PXRD characterization of the recovered catalyst (Figure 5) shows that [BMIM]_3_PMo_12_@ZIF-8 undergoes structural transformations during its catalytic use, evidenced by clear differences between the diffraction patterns of the catalyst before and after being subjected to the ODS/ODN experiments. Interestingly, a considerable structural transformation is verified from the first catalytic cycle, where the typical crystalline phase of the ZIF-8 changes, suggesting that the support materials suffer structural modification from the beginning of the reaction. Furthermore, the results obtained by ^31^P NMR showed no signal attributable to [BMIM]_3_PMo_12_, which suggests that there was no leaching of the [BMIM]_3_PMo_12_ active center.

The SEM images obtained after the first and the tenth ODS/ODN cycles are present in Figure 6a. After the first cycle the composite still presents particles with regular rhombic dodecahedron morphological regularity as observed in the composite before catalytic use. However, after the tenth ODS/ODN cycle, the composite presents a not well defined morphology without uniform particles. This result corroborates with the previous XRD data also performed for this tenth cycle (Figure 6b). Nevertheless, EDS elemental mappings and spectra (Figure 5) still confirm the maintenance of a uniform distribution of the main elements of the composite even after the tenth cycle, namely Molybdenum, Phosphorus (from the [BMIM]_3_PMo_12_) and Zinc (from de ZIF-8). EDS dates were further used as Zn/Mo ratio to compare the content for the initial composite and for the composite recovered after the first and the tenth catalytic cycles. The ratio Zn/Mo increased from 1.32 to 2.91 from the initial composite to the same recovered after the first catalytic cycle, and consequently to 3.7 after the tenth cycle. These results indicate the occurrence of Mo leaching during the reutilization process. However, the active [BMIM]_3_PMo_12_ was not detected by ^31^P NMR analysis, probably due to the decomposition of Keggin polyoxomolybdate structures during the catalytic reaction. This POM decomposition probably forms smaller sized polyoxomolybdate fragments that can leach from the ZIF-8 structure.

A scarce number of reported works can be found studying a simultaneously oxidative desulfurization and oxidative denitrogenation of fuels using heterogeneous catalysts. Aghbolagh et al. used a copper based catalyst TBAPMo_11_Cu@CuO that presented high efficiency (>90%) for S and N oxidation; however, low efficiency for the extraction of oxidized products was found using an organic inadequate extraction solvent [44]. Other work presented by Ishihara et al. used MoOx impregnated into aluminium oxide as a catalyst. In this case a high catalytic efficiency was found using tert-butyl hydroperoxide as an oxidant [45]. This oxidant presents a lower viability to be used in an industrial process and, also, the recycle capacity of the catalyst was not discussed. The work reported by Palomeque-Santiago using WOx on ZrO_2_ still presents an ODS/ODN efficiency of 97%; however, the recycle capacity and the stability of the catalyst was not presented in this study either [14]. Recently, our research group has been reporting the application of PMo_12_@MOF-808 as a catalyst for simultaneous ODS/ODN processes [30]. While showing a high catalytic efficiency, this catalyst decreases its activity after the first three or four catalytic cycles. Compared with these few published works, the [BMIM]_3_PMo_12_@ZIF.8 catalyst presents identical high efficiency and high recycle capacity, although its structural transformation is probably necessary to maintain its activity for at least 10 consecutive ODS/ODN cycles.

## 3. Experimental section

### 3.1. Materials and Methods

Phosphomolybdic acid (H_3_PMo_12_O_40_·xH_2_O, for microscopy), 2-methylimidazole (C_4_H_6_N_2_, 99.0%, 2-MIM), tetradecane (C_14_H_30_, >99.0%), 1-butyl-3-methylimidazole hexafluorophosphate (C_8_H_15_F_6_N_2_P, >97%, [BMIM]PF_6_), hydrogen peroxide (H_2_O_2_, aq. 30%), dibenzothiophene (C_12_H_8_S, 98%, DBT) and 4-methyldibenzothiophene (C_13_H_10_S, 96%, MDBT), quinoline (98%, QUI) and indole (> 99%, IND) were obtained from Sigma-Aldrich. *N*,*N*-dimethylformamide (C_3_H_7_NO, 99.99%, DMF), methanol (CH_3_OH, analytical reagent grade) and ethanol (C_2_H_5_OH, >99.8%, EtOH) were purchased from Fisher. Acetonitrile (CH_3_CN, >99.5%, MeCN), 1-butyl-3-methylimidazolium bromide (C_8_H_15_N_2_Br, >97%, [BMIM]Br) and benzothiophene (C_8_H_6_S, >95%, BT) were acquired from Fluka. Zinc(II) nitrate tetrahydrate (Zn(NO_3_)_2_·4H_2_O, >98.5%) was obtained from Merck. N-octane (C_8_H_18_, >99.0%) and 4,6-dimethyldibenzothiophene (C_14_H_12_S, 95%, DMDBT) were acquired from Acros Organic. None of these was the subject of further treatment or purification.

Fourier-transform infrared (FT-IR) spectra were acquired on the attenuated total reflectance (ATR) operation mode of a PerkinElmer FT-IR System Spectrum BX spectrometer, and all the representations are shown in arbitrary unities of transmittance. Elemental analyses (CHN) were performed using a ThermofinniganFlash EA 112 series (University of Santiago de Compostela, Spain). Powder X-ray diffraction (PXRD) patterns were obtained at room temperature on a Rigaku Geigerflex diffractometer operating with a Cu radiation source (*λ*_1_ = 1.540598 Å; *λ*_2_ = 1.544426 Å; *λ*_1_/*λ*_2_ = 0.500) and in a Bragg-Brentano *θ*/2*θ* configuration (45 kV, 40 mA). Intensity data were collected by a step-counting method (step 0.026°), in continuous mode, in the 3 ≤ 2*θ* ≤ 50 ° range, and all the re-presentations are shown in arbitrary unities of intensity. Scanning electron microscopy (SEM) and electron dispersive X-ray spectroscopy (EDS) analysis were performed in a FEI Quanta 400 FEG ESEM high resolution scanning electron microscope equipped with an EDAX Genesis X4M spectrometer working at 15 kV. Samples were coated with a Au/Pd thin film by sputtering using a SPI Module Sputter Coater equipment. ^31^P nuclear magnetic resonance (^31^P NMR) spectra were acquired in CD_3_CN at 162 MHz with a Bruker Avance III 400 spectrometer, and the chemical shifts are given with respect to external 85% H_3_PO_4_. Inductively coupled plasma optical emission spectroscopy (ICP-OES) was used to quantify the Mo concentration, resorting to a PerkinElmer Otima 4300 DV. Catalytic reactions were periodically monitored by GC-FID analysis carried out in a Bruker 430-GC-FID chromatograph. Hydrogen was used as carrier gas (55 cm·s^−1^) and fused silica Supelco capillary columns SPB-5 (30 m × 0.25 mm i. d.; 25 μm film thickness) were used.

### 3.2. Materials Preparation

#### 3.2.1. [BMIM]_3_PMo_12_

This phosphomolybdate salt was prepared following a procedure adapted from a method previously reported [17]. [BMIM]Br (2.5 mmol) was dissolved in 2.5 mL of deionized water. To this, an aqueous solution of phosphomolybdic acid (0.5 mmol) in deionized water (2.5 mL) was added dropwise under magnetic stirring at room temperature. The mixture was left stirring for 30 min after which the precipitate was filtrated, washed thoroughly with deionized water and dried in a desiccator overnight. FTIR (cm^−1^): *ν*([BMIM]_3_PMo_12_) = 3150 (w), 3098 (w), 2965 (w), 1603 (vw), 1564 (w), 1463 (w), 1383 (vw), 1338 (vw), 1250 (vw), 1164 (m), 1060 (s), 953 (s), 872 (m), 781 (s), 744 (s), 619 (m), 500 (m), 463 (m). ^31^P NMR (162 MHz, CD_3_CN, 25 °C) δ = −2.38 ppm. Anal.Calc. for (C_8_H_15_N_2_)_3_PMo_12_O_40_(2239.88): C, 13.20; H, 1.99; N, 3.72; Found: C,13.19; H, 2.01; N, 3.72.

#### 3.2.2. ZIF-8 and [BMIM]_3_PMo_12_@ZIF-8

These materials were prepared following an experimental procedure adapted from previously reported methods [24]. For the ZIF-8, an initial solution of Zn(NO_3_)_2_·4H_2_O (2.5 mmol) in MeOH (25 mL) was magnetically stirred for 15 min. Then, a solution of 2-MIM (19.8 mmol) in MeOH (25 mL) was slowly added. The resulting reactional mixture was magnetically stirred for 2.5 h, at ambient temperature, after which the resulting material was isolated by centrifugation, washed five times with MeOH and dried at 60 °C under vacuum overnight. For [BMIM]_3_PMo_12_@ZIF-8, a solution of [BMIM]_3_PMo_12_ (100 mg) in DMF (5 mL) was added to the first one and the mixture was left to stir for 15 min before the addition of the 2-MIM solution.

### 3.3. Oxidative Desulfurization and Denitrogenation (ODS/ODN)

ODS and ODN studies were performed with a model fuel prepared with a group of compounds representative of the most refractory sulfur and nitrogen content in diesel, namely benzothiophene (BT), dibenzothiophene (DBT), 4-methyldibenzothiophene (MDBT), 4,6-dimethyldibenzo-thiophene (DMDBT), quinoline (QUI) and indole (IND), dissolved in n-octane (500 ppm of sulfur or 300 ppm of nitrogen from each compound). Aqueous hydrogen peroxide was used as an oxidizing agent. The reactions were carried out under air in a closed borosilicate vessel with a magnetic stirrer and immersed in a thermostatically controlled liquid paraffin bath at 70 °C. ODS/ODN reactions were performed in a biphasic system composed by the model fuel and [BMIM]PF_6_ as extraction solvent with ratio 1:1 (*v*/*v*). The simultaneous desulfurization and denitrogenation experiments were performed in two main separated steps; initially an extraction of sulfur and nitrogen compounds was transferred from model fuel to the extraction phase just by stirring the biphasic system. Then, the oxidative catalytic step was initiated by adding the oxidant. In a representative experiment, a certain amount of heterogeneous catalyst equivalent to 3 mmol of active POM ([BMIM]PMo_12_) was added to 0.75 mL of [BMIM]PF_6_ and 0.75 mL of model fuel. This mixture was stirred for 10 min at 70 °C. The oxidative catalytic step was then initiated with the addition of aqueous H_2_O_2_ 30% (75 μL) to the reaction mixture. Tetradecane was used as a standard in the periodical monitorization of the sulfur content by GC analysis. At the end of each reaction, the model fuel phase was removed and substituted with an equal volume of fresh model fuel, after which a subsequent catalytic cycle was started, under identical reaction conditions. After 10 catalytic cycles, the catalyst was recovered by centrifugation, washed carefully for three times with MeCN and EtOH, and dried at 60 °C under vacuum overnight.

## 4. Conclusions

A novel composite prepared by the encapsulation of the ionic liquid polyoxomolybdate [BMIM]_3_PMo_12_ into a ZIF-8 framework was successfully isolated and characterized. Furthermore, the isolated material [BMIM]_3_PMo_12_@ZIF-8 was used as heterogeneous catalyst to simultaneously remove sulfur and nitrogen from a model fuel via oxidation. In a competitive oxidative environment, this catalyst showed high efficiency and the absence of active centre leaching during ten consecutive model fuel treatments. Complete desulfurization and denitrogenation were achieved after only one hour, which is comparable with the catalytic performance of the corresponding homogeneous catalyst, [BMIM]_3_PMo_12_. Remarkably, the structural transformation of the [BMIM]_3_PMo_12_@ZIF-8 observed from the first catalytic cycle seems too crucial for the high catalytic performance of this solid, in particular for the preservation of the catalytic activity during the ten reaction cycles. Despite the interesting result reported, further systematic studies will be needed to achieve a more complete understanding of the nature of the structural transformation.

## Data Availability

More data can be obtained by request to the authors.

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
