# Peer review of "Ionic Liquid-Based Polyoxometalate Incorporated at ZIF-8: A Sustainable Catalyst to Combine Desulfurization and Denitrogenation Processes"

_molecules, 2022, doi:10.3390/molecules27051711_

Round 1

Reviewer 1 Report

The authors reported a novel heterogeneous catalyst by the encapsulation of an imidazolium-based polyoxometalate (POM) into ZIF-8 framework ([BMIM]PMo12@ZIF-8). This composite material revealed exceptional catalytic efficiency to simultaneous proceed the oxidative desulfurization and denitrogenation of a multicomponent model fuel containing various sulfur and nitrogen compounds. However, there are some problems that need to be corrected. I suggest that it be accepted with minor modifications.

  1. What is the difference between [BMIM]3PMo12@ZIF-8 and [BMIM]3PMo12@ZIF-8(AC), what is the meaning of “AC”. Why the XRD of these two samples are so different, more detailed analysis of XRD patterns is required. In addition, the XRD of [BMIM]3PMo12 should be provided.
  2. The first paragraph of section 2.2, an introduction to catalytic experiments, should be moved to section “3. Experimental section”.
  3. The authors provided some EDS date as shown in Figure 2 and5, but not the concentration of each element.
  4. Only comparison of the distribution of elements before and after the reaction cannot ensure the stability of the catalysts, the comparison of the composition and structural characteristics before and after the reaction is more important.
  5. In the desulfurization and denitrogenation experiments, the catalytic activities of ZIF-8 and control samples should also be provided.
  6. Why is the catalytic performance of heterogeneous catalyst higher than [BMIM]3PMo12, and the corresponding promotion mechanism needs to be clarified.

Author Response

Reviewer 1:

The authors reported a novel heterogeneous catalyst by the encapsulation of an imidazolium-based polyoxometalate (POM) into ZIF-8 framework ([BMIM]PMo12@ZIF-8). This composite material revealed exceptional catalytic efficiency to simultaneous proceed the oxidative desulfurization and denitrogenation of a multicomponent model fuel containing various sulfur and nitrogen compounds. However, there are some problems that need to be corrected. I suggest that it be accepted with minor modifications.

  1. What is the difference between [BMIM]3PMo12@ZIF-8 and [BMIM]3PMo12@ZIF-8(AC), what is the meaning of “AC”. Why the XRD of these two samples are so different, more detailed analysis of XRD patterns is required. In addition, the XRD of [BMIM]3PMo12 should be provided.

Response: The authors acknowledge the reviewer for this point. The designation [BMIM]3PMo12@ZIF-8 is for the composite before catalytic use and the designation [BMIM]3PMo12@ZIF-8(AC) corresponds to the same composite after used for consecutive ODS/ODN cycles. The XRD before and after ten catalytic cycles is different, most probably caused by some ZIF-8 structural alteration after the ten catalytic cycles. This modification is already observed after the first catalytic cycle, and is most likely caused by the catalytic reaction, occurring a structural modification of the composite by the presence of oxidant and the reactants and products from reaction.

The XRD of the active POM [BMIM]3PMO12 was incorporated in Fig. 1a. The XRD pattern of the POM is different of the initial composite and also to the composite after catalytic use (Fig 5). This indicate that the POM is distributed in to the ZIF-8 porous framework.

  1. The first paragraph of section 2.2, an introduction to catalytic experiments, should be moved to section “3. Experimental section”.

Response: Most of the information presented in the first paragraph of section 2.2 was transferred to experimental section 3.3. All the modifications in the manuscript were highlighted at green. 

  1. The authors provided some EDS date as shown in Figure 2 and 5, but not the concentration of each element.

Response: The authors acknowledge the reviewer for this point. Initially, EDS technique were used mainly to identify the elements. The quantification of elements by EDS does not present high accuracy. Therefore, EDS date were used at this moment as the ratio of elements between ZIF-8 (Zn) and the PMo12 (Mo). Quantification Zn/Mo ratio were performed for the initial composite [BMIM]3PMo12@ZIF-8, for the composite recovered after the first catalytic cycle [BMIM]3PMo12@ZIF-8(AC1) and after the 10th cycle ([BMIM]3PMo12@ZIF-8(AC10). The ratio Zn/Mo increased from 1.32 to 2.91 from the initial to the [BMIM]3PMo12@ZIF-8(AC1), and consequently to 3.7 after the 10h cycle. These results suggest the occurrence of Mo leaching during the reutilization process. However, the active [BMIM]3PMo12 was not detected by 31P NMR analysis, probably due to the decomposition of the Keggin polyoxomolybdate structure during the catalytic reaction. This POM decomposition probably forms smaller sized polyoxomolybdate fragments that can leach from the ZIF-8 structure. 

This information was introduced in section 2.2 and it is highlighted at green.

  1. Only comparison of the distribution of elements before and after the reaction cannot ensure the stability of the catalysts, the comparison of the composition and structural characteristics before and after the reaction is more important.

Response: The authors agree with the reviewer and acknowledge this opportunity to improve the discussion about the stability of the catalyst. To improve this part, SEM image after the first cycle was introduced, as well as the XRD analysis. The ratio of Zn/Mo corresponding to the composite before and after catalytic use was also quantified and the data was incorporated in the section 2.2 in the manuscript.

  1. In the desulfurization and denitrogenation experiments, the catalytic activities of ZIF-8 and control samples should also be provided.

Response: The authors agree with the reviewer and acknowledge for this point. Experiment using the ZIF-8 instead of the composite [BMIM]3PMo12@ZIF-8 was performed to assess its inherent catalytic performance; however, ZIF-8 did not present any catalytic activity. This information was added in section 2.2 and it is highlighted at green.

  1. Why is the catalytic performance of heterogeneous catalyst higher than [BMIM]3PMo12, and the corresponding promotion mechanism needs to be clarified.

Response: This is an important question. The application of the homogeneous polyoxomolybdate PMo12 catalyst has been studied before and published recently by our research group (Sustain. Chem. 2021, 2, 382-391, ref [27]). In this previous work, although the high efficiency found, structural transformation of PMo12 was found that led to its deactivation. In the case of heterogeneous catalyst, the [BMIM]3PMo12 is immobilized in the porous of ZIF-8, what can promote some POM structural stabilization, what it will originate a slightly higher catalytic performance.

The mechanism in this POMs/H2O2 systems is well reported in the literature. POM active center activates the oxidant by the interaction of a terminal or a bridged oxygen from the Keggin structure with oxidant that promoted the formation of a peroxometalate. This last will interact with the reactant promoting the formation of sulfoxide, followed in the next oxidation step to sulfone (or nitrogen oxides for ODN). In the present work, a structural transformation of the catalyst occurred, probably to transform the initial composite [BMIM]3PMo12@ZIF-8 in to a more stable and still high efficient catalyst. To present a correct mechanism in this case, exhaustive structural studies are need. The authors intend to perform this study in a near future.  

Reviewer 2 Report

This paper presents preparation of assembled polyoxometalate and ZIF-8 and its use as the catalyst for desulfurization and denitrogenation of polyaromatic compounds containing N or S atom. Results of this study included SEM characterization of the catalyst and their reutilization performance.

I have a question about the IR spectra of ZIF used in this study. A sentence from line 82 to 87 and Figure 1(b) indicated that the IR spectrum of ZIF showed a strong absorption due to Zn-N vibration at 420 cm-1 and weak C=N vibration peak at 1580 cm-1. Previous papers on this subject, such as ref 41, presented the IR peak due to Zn-N stretching vibration in much weaker intensity than the peaks assigned to the organic groups. In general, the coordination bonds, such as M-N, M-O, and M-S bonds, show weaker IR peaks than the vibrations of organic molecules. Is this due to a much lower content of the imidazole group of ZIF-8 used in this study than ZIF in other studies?

Author Response

Reviewer 2:

This paper presents preparation of assembled polyoxometalate and ZIF-8 and its use as the catalyst for desulfurization and denitrogenation of polyaromatic compounds containing N or S atom. Results of this study included SEM characterization of the catalyst and their reutilization performance.

I have a question about the IR spectra of ZIF used in this study. A sentence from line 82 to 87 and Figure 1(b) indicated that the IR spectrum of ZIF showed a strong absorption due to Zn-N vibration at 420 cm-1 and weak C=N vibration peak at 1580 cm-1. Previous papers on this subject, such as ref 41, presented the IR peak due to Zn-N stretching vibration in much weaker intensity than the peaks assigned to the organic groups. In general, the coordination bonds, such as M-N, M-O, and M-S bonds, show weaker IR peaks than the vibrations of organic molecules. Is this due to a much lower content of the imidazole group of ZIF-8 used in this study than ZIF in other studies?

Response: The powder XRD patterns unequivocally confirm the successful preparation of the ZIF-8 materials in the expected crystalline phase, i. e. with the usual reason between the imidazole ligands and the metal centers. Consequently, the strong absorption peak observed in the reported FTIR spectrum due to Zn-N vibration (at 420 cm-1) relatively to other weak peaks assigned to organic groups is not consequence of a lower content of the imidazole groups of ZIF-8 in this study compared with other studies. In fact, the high relative intensity of the Zn-N vibration peaks is related with the acquisition of the Fourier-Transformed Infrared (FTIR) spectra on the attenuated total reflectance (ATR) operation mode, instead the transmission mode (used in the reference 41). There are several reported examples of ATR-FTIR spectra of the ZIF-8 material showing the Zn-N absorption peak with higher intensity than the other peaks, for example: ACS Appl. Energy Mater. 2020, 3, 2925−2934 (doi: dx.doi.org/10.1021/acsaem.0c00009) from our research group; Cryst. Growth Des. 2019, 19, 4844−4853 (doi: 10.1021/acs.cgd.9b00838); J. Mater. Sci. 2020, 55, 6130–6144 (doi: 10.1007/s10853-020-04452-6); RSC Adv. 2021, 11, 39169-39176 (doi: 10.1039/d1ra07089d).

Reviewer 3 Report

Article is well written and presented. However, it is quite surprising that inexplicable structural transition occurs after several cycles of the composite use. This fact gives me a very important question - which substance really possesses the desulfurization and denitrogenation properties which are in the main focus of the paper. Thus, some more data on characterization of the prepared catalyst is needed. 

1. At which cycle the phase transition occurs? PXRD of the composite should be presented after each cycle. 

2. It is possible to elucidate the structure of the new phase if it is a known phase. If ZIF-8 framework decomposes while treated by ionic liquid, there might be just some insoluble inorganic POM/oxide phase. 

3. Elemental analysis and EDX data for the composite should be presented  after each cycle. Furthermore, Zn/P/Mo ratio should be determined for the as-synthesized composite to show the real content of POM in the catalyst. 

4. Finally, I doubt the utility of the prepared heterogeneous catalyst showing 80-90% efficiency. Some more previously reported ODS/ODN catalysts should be compared to the [BMIM]3PMo12@ZIF-8 and discussed to disclose more brightly its real applicability. 

After solving these stated problems, the main text seemingly needs to be vastly reconsidered. 

Author Response

Article is well written and presented. However, it is quite surprising that inexplicable structural transition occurs after several cycles of the composite use. This fact gives me a very important question - which substance really possesses the desulfurization and denitrogenation properties which are in the main focus of the paper. Thus, some more data on characterization of the prepared catalyst is needed. 

  1. At which cycle the phase transition occurs? PXRD of the composite should be presented after each cycle. 

Response: The authors acknowledge the reviewer for this point. To better investigate the stability of the catalyst XRD and SEM/EDS were performed after the first and the tenth cycle. Unfortunately, it was not possible for us to perform this study after each ODS/ODN cycle because the time limitation of equipment use and also the amount of fresh catalyst available at the moment. The XRD analysis performed after the first cycle already demonstrate the occurrence of structural modification of composite. By the SEM/EDS analysis this transformation is not clear after the first cycle but after the tenth cycle the complete structural alteration is well identified. A clearer and complete discussion about the stability of the catalyst of incorporated in the manuscript and it is highlighted at green in section 2.2 and in figures 5 and 6.

  1. It is possible to elucidate the structure of the new phase if it is a known phase. If ZIF-8 framework decomposes while treated by ionic liquid, there might be just some insoluble inorganic POM/oxide phase. 

Response: This is in fact an important question. Unfortunately, the XRD pattern observed after the first and the tenth cycles are not similar to the PMo12 (Figure 1a) or MoO2 NPs (several examples in the literature). In case of insoluble POM/oxide phase, this could correspond to the occurrence of ZIF-8 dissolution and complete destruction. This is not observed in our work, confirmed by the identification and quantification of Zn elements by EDS. Therefore, this structural modification caused by the catalytic reaction must corresponds to MOF structural modification.

Further, catalytic experiments were performed using the isolated support ZIF-8 and in this case no catalytic activity was found. Further, to analyse the stability of the ZIF-8, XRD analysis were performed after catalytic use and in this case the diffractogram pattern was maintained (results do not showed in this manuscript), what indicates a structural maintenance of the MOF in the ionic liquid solution when oxidative catalysis did not occurred. The structural modification observed with the composite is related with its catalytic performance.

  1. Elemental analysis and EDX data for the composite should be presented  after each cycle. Furthermore, Zn/P/Mo ratio should be determined for the as-synthesized composite to show the real content of POM in the catalyst. 

Response: The authors agree with the reviewer and acknowledge for this point. As previously mentioned, it was not possible for us to perform the SEM/EDS study after each ODS/ODN cycle because the time limitation of equipment use and also the amount of fresh catalyst available at the moment. SEM/EDS and EDS quantification was performed after the first and the tenth cycles. Quantification Zn/Mo ratio were performed for the initial composite [BMIM]3PMo12@ZIF-8, for the composite recovered after the first catalytic cycle [BMIM]3PMo12@ZIF-8(AC1) and after the 10th cycle ([BMIM]3PMo12@ZIF-8(AC10). The ratio Zn/Mo increased from 1.32 to 2.91 from the initial to the [BMIM]3PMo12@ZIF-8(AC1), and consequently to 3.7 after the tenth cycle. These results indicate the occurrence of Mo leaching during the reutilization process. However, the active [BMIM]3PMo12 was not detected by 31P NMR analysis, probably due to the decomposition of the Keggin polyoxomolybdate structure during the catalytic reaction. This POM decomposition probably forms soluble smaller sized polyoxomolybdate fragments that can leach from the ZIF-8 structure. This information was introduced in section 2.2 and it is highlighted at green.

  1. Finally, I doubt the utility of the prepared heterogeneous catalyst showing 80-90% efficiency. Some more previously reported ODS/ODN catalysts should be compared to the [BMIM]3PMo12@ZIF-8 and discussed to disclose more brightly its real applicability. 

Response: The authors agree with the reviewer and acknowledge for this point. The comparison of the results obtained with other published works is essential. In fact, works reporting simultaneously oxidative desulfurization and oxidative denitrogenation of fuels using heterogeneous catalysts is still scarce and recent in the literature. Apart from our work reporting the application of PMo12@MOF-808  catalyst in simultaneous ODS/ODN processes (referenced in the manuscript as [30]), few works can be find using POMs based heterogeneous catalysts in ODS/ODN. Aghbolaghet al. used a copper based catalyst TBAPMo11Cu@CuO and high efficiency (> 90%) for S and N oxidation were achieved. However, low extraction of oxidized products has been find using organic inadequate extraction solvent (Energy Fuels 2020, 34, 12, 16366–16380). Other more recent publications were using MoOx and WOx active centers impregnated in aluminium oxide and zirconia, respectively. The first work, presented by Ishihara et al. used TBHP instead of H2O2, that presents a lower viability to be used in an industrial process. Also, the recycle capacity of the catalyst is not presented (Applied Catalysis A: General, 279, 2005, 279-287). The work reported by Palomeque-Santiago using WOx on ZrO2 (Applied catalysis B: Environ, 236, 2018, 326-337) still present ODS/ODN efficiency of 97%; however, also in this study the recycle capacity and the stability of the catalyst is not discussed. According to the limited number of studies reporting simultaneous ODS/ODN studies and the absence of catalyst recycling and stability analysis, the present manuscript presents a catalyst with high efficiency, high recycle capacity although its structural transformation probably necessary to maintain its activity for at least 10 consecutive ODS/ODN cycles. This information was introduced in the manuscript in section 2.2, after line 172.

Round 2

Reviewer 2 Report

I recommend publication of this version of the manI recommend publication of this version of the manuscript.

Reviewer 3 Report

Authors have done all that was possible to improve the manuscript and the results clarity, and now the paper can be accepted for publication.